# A *Catharanthus roseus* Fe(II)/α-ketoglutarate-dependent dioxygenase catalyzes a redox-neutral reaction responsible for vindolinine biosynthesis

Jasmine Ga May Eng[1], Mohammadamin Shahsavarani[2], Daniel Patrick Smith[1], Josef Hájíček[3], Vincenzo De Luca [4] & Yang Qu [1,2 ✉]

The Madagascar's periwinkle is the model plant for studies of plant specialized metabolism and monoterpenoid indole alkaloids (MIAs), and an important source for the anticancer medicine vinblastine. The elucidation of entire 28-step biosynthesis of vinblastine allowed further investigations for the formation of other remarkably complex bioactive MIAs. In this study, we describe the discovery and characterization of vindolinine synthase, a Fe(II)/α-ketoglutarate-dependent (Fe/2OG) dioxygenase, that diverts assembly of tabersonine to vinblastine toward the formation of three alternatively cyclized MIAs: 19*S*-vindolinine, 19*R*-vindolinine, and venalstonine. Vindolinine synthase catalyzes a highly unusual, redox-neutral reaction to form a radical from dehydrosecodine, which is further cyclized by hydrolase 2 to form the three MIA isomers. We further show the biosynthesis of vindolinine epimers from tabersonine using hydrolase 2 catalyzed reverse cycloaddition. While the occurrence of vindolinines is rare in nature, the more widely found venalstonine derivatives are likely formed from similar redox-neutral reactions by homologous Fe/2OG dioxygenases.

[1] Department of Chemistry, University of New Brunswick, Fredericton, NB, Canada. [2] Department of Chemical Engineering, University of New Brunswick, Fredericton, NB, Canada. [3] Department of Chemistry, Charles University in Prague, Praha, Czech Republic. [4] Department of Biological Sciences, Brock University, St. Catharines, ON, Canada. ✉email: yang.qu@unb.ca

onoterpenoid indole alkaloids (MIAs) are among the most complex and diverse alkaloids in nature with more than 3000 reported structures[1,2]. These impressive natural products offer a vast array of bioactivities, exemplified by the anticancer drugs vinblastine and camptothecin, the antihypertensive ajmalicine, and the analgesic mitragynine. Among the MIA producing species, *Catharanthus roseus* (Madagascar's periwinkle) has become the model plant for studying specialized MIA metabolism with extensive studies on the biochemistry, MIA pathway regulation, cellular compartmentation, and lately metabolic engineering of MIAs. Recently, the complete 28-step biosynthesis of vinblastine has been fully characterized, illustrating a remarkably complicated yet highly coordinated biosynthetic pathway and laying the foundation for the discovery of other medicinal MIA pathways and their engineering[3–12].

The formation of vinblastine requires coupling of the two most abundant *C. roseus* MIAs catharanthine (iboga type) and vindoline (aspidosperma type) by a peroxidase[13]. The spatial separation of these two MIAs is likely a reason for the low abundance of vinblastine in the plants[14]. While the assembly of catharanthine and vindoline has been fully elucidated, biosynthesis of the 3rd most abundant MIAs in *C. roseus*, namely vindolinine (19 R) and its diastereomer epi-vindolinine (19 S) (Fig. 1), have yet to be elucidated. In this study, we describe the discovery and characterization of vindolinine synthase (VNS), a Fe(II)/α-ketoglutarate-dependent (Fe/2OG) dioxygenase responsible for the formation of both epimers. Using the co-substrates α-ketoglutarate, O₂, and the highly unstable intermediate dehydrosecodine generated by the dihydroprecondylocarpine acetate synthase (DPAS), or geissoschizine synthase (GS), VNS catalyzes an unusual, redox-neutral reaction to generate a radical intermediate that is further cyclized by *C. roseus* hydrolase 2/tabersonine synthase (HL2/TabS) or hydrolase 4 (HL4) to generate the two diastereomeric vindolinines and a third isomer venalstonine that accumulate in various ratios in *C. roseus*. This discovery highlights the malleability and plasticity of MIA biosynthesis, one of the key drivers for the MIA diversity, and showcases the reaction versatility of Fe/2OG dioxygenases by presenting a rare, redox neutral reaction with respect to the MIA substrate and product formed.

## Results

**Vindolinines are the third most abundant MIAs in *C. roseus* leaves next to vindoline and catharanthine.** Previous results showed that MIAs are selectively extracted by dipping intact *C. roseus* leaves in chloroform followed by extracting the remaining MIAs by submerging leaves in methanol[14]. This simple technique effectively extracted catharanthine, 19S-vindolinine, 19R-vindolinine, perivine and ajmalicine in the chloroform, while most of the vindoline, vindorosine (demethoxyvindoline), and serpentine were extracted after submitting chloroform treated leaves with methanol. The identity of the two vindolinine epimers were confirmed by comparing to a commercial standard for 19R-vindolinine and Nuclear Magnetic Resonance (NMR) analyses of 19S-vindolinine purified from *C. roseus* leaf chloroform extract (Supplementary Fig. 1, Supplementary Table 1)[15,16]. The vindolinines are structurally related with the aspidosperma type MIA tabersonine, but differ by the addition of C2-C19 bridge and reduction of the C2-C16 double bonds (Fig. 1). They also differ from each other by the C19 stereochemistry, resulting in 0.4 ppm chemical shift difference for H18 (Supplementary Fig. 1)[15,16]. In addition to the two vindolinines, we were also able to detect and purify venalstonine from the same chloroform extract and confirmed its identity by NMR (Supplementary Fig. 2, Supplementary Table 1)[17,18] and liquid chromatography-tandem mass spectrometry (LC-MS/MS) (Supplementary Fig. 3). Compared to the vindolinines, venalstonine instead contains a C2-C18 bridge that lacks a stereocenter on C18 or C19. The MS/MS product ion scans showed similar fragmentation pattens, as well as a signature daughter ion of *m/z* 320 (loss of O⁻) unique to the vindolinine epimers and venalstonine (Supplementary Fig. 3). All three MIAs also showed typical UV absorption spectra as anticipated from a dihydroindole chromophore (Supplementary Fig. 3).

By extracting whole ground tissues with methanol, the major MIAs in *C. roseus* leaves and flowers were quantified using a number of standards in the plant variety "Little Delicata" and in another popular variety "Pacifica White". The results showed that while the vindolinine epimers were the third-most abundant MIAs in leaves of both varieties next to vindoline and catharanthine, they were the most abundant MIAs in flowers of Little Delicata and the second most abundant after vindoline/vindorosine in flowers of Pacifica White (Supplementary Table 2).

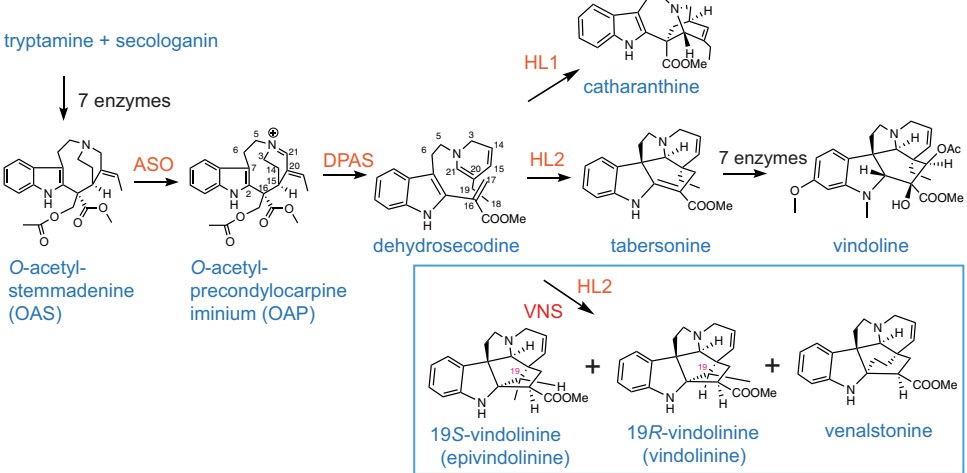

**Fig. 1 The biosynthetic pathways from *O*-acetylstemmadenine for the three most abundant MIAs in *Catharanthus roseus* leaves: catharanthine, vindoline, and vindolinine epimers (19*R*-vindolinine, 19*S*-vindolinine) as well as venalstonine.** ASO: *O*-acetylstemmadenine oxidase, DPAS: dihydroprecondylocarpine acetate synthase, HL1: hydrolase 1, HL2: hydrolase 2, VNS: vindolinine synthase. HL1 and HL2 have also been named catharanthine synthase and tabersonine synthase, respectively.

**Identification of the Fe(II)/α-ketoglutarate-dependent dioxygenase VNS in _C. roseus_ leaf epidermal transcriptome.** The biosynthesis of MIAs in _C. roseus_ leaves is largely located in the leaf epidermis[6,12,19,20]. Loganic acid produced in the leaf internal phloem associated parenchyma (IPAP) cells is transported to the leaf epidermis, where it is further _O_-methylated and oxidatively cleaved to form the key iridoid secologanin[21,22]. The secologanin is coupled to tryptamine, generated by decarboxylation of tryptophan, to form strictosidine, the central precursor to almost all 3,000 MIAs. A number of enzymes are responsible for converting strictosidine to two late intermediates catharanthine and tabersonine in the leaf epidermis, where the latter continues to be decorated to form desacetoxyvindoline before it is exported to leaf laticifers and idioblasts. The last two steps of vindoline formation in these two cell types requires an oxidation by desacetoxyvindoline 4-hydroxylase (D4H), the only known Fe/2OG dioxygenase in all MIA biosynthesis and 4-_O_-acetylation catalyzed by deacetylvindoline _O_-acetyltransferase (DAT)[23,24].

The structural connection between vindolinines and tabersonine and their presence in the leaf chloroform extract suggest the biosynthesis is likely located in leaf epidermis. The occurrence of vindolinines is rare in nature and they are only reported in two other Apocynaceae species, _Vinca erecta_ and _Melodinus balansae_[25]. This suggests that the enzyme(s) involved must not be commonly shared among other MIA producing species. It is possible that the unstable dehydrosecodine intermediate generated during the ring-opening of _O_-acetylprecondylocarpine (OAP) required for the formation of tabersonine and catharanthine (Fig. 1) may also be a precursor for the biosynthesis of vindolinines[26]. With this information, we searched genes from a _C. roseus_ leaf epidermal dataset previously used to identify a number of MIA biosynthetic genes[4–6,19,20,27]. Specifically, we looked for genes that are unique to _C. roseus_ and not found in three other MIA producing species _Amsonia hubrichtii_, _Vinca minor_ and _Tabernaemontana elegans_ that produce aspidosperma MIAs but not vindolinines[28]. This search resulted in the identification of a Fe/2OG dioxygenase VNS (Genbank OL677442) that is highly represented in the _C. roseus_ leaf epidermis dataset [Expressed Sequence Tag (EST) number of 83] while showing no amino acid sequence identity of more than 50% to dioxygenases in the other three species. For comparison, other known epidermal vinblastine pathway enzymes secologanin synthase (SLS) and geissoschizine oxidase (GO) have relative EST numbers, a rough reflection of gene expression levels, of 54 and 38, respectively in this dataset. Quantitative Reverse Transcription PCR (qRT-PCR) supported that VNS expression is enriched in leaf epidermis total RNAs extracted by the carborundum abrasion technique[20] compared to those from whole leaf tissues, which was also detected in all organs of _C. roseus_ including flowers, roots, and stems (Supplementary Fig. 4). Sequence alignment of VNS with other known Fe/2OG dioxygenases showed that VNS contains the HXDXnH catalytic facial triad required for coordinating Fe(II) and the YXnRXS motif involved in α-ketoglutarate binding in these enzymes (Supplementary Fig. 4)[29]. In a phylogeny analysis, the mostly related gene is a Fe/2OG dioxygenase from _Vinca minor_ with 48% amino acid identity (Supplementary Fig. 5).

**Virus induced gene silencing (VIGS) confirmed VNS involvement in vindolinine biosynthesis.** Next, we performed VIGS experiments to investigate the in vivo VNS function in _C. roseus_. The leaves of VNS-silenced plants contained 95% lower levels of 19_R_- and 19_S_-vindolinine compared to empty vector (EV) controls, as a result of silencing VNS transcript levels by 93% (Fig. 2a–c). A small reduction and increase of catharanthine and

vindoline levels, respectively, were observed in VIGS plants compared to the EV controls (Fig. 2a). While these small changes may not be easily explained without further analyses, the drastic reduction of the vindolinine levels strongly suggested that VNS is involved in 19 _S/R_-vindolinine production _in planta_.

**Vindolinines are formed from the same dehydrosecodine intermediate to the iboga and aspidosperma MIAs.** To further investigate VNS function, we transiently expressed VNS and other MIA biosynthetic genes in _Nicotiana benthamiana_ (tobacco) leaves, which provided required enzyme co-factors and facilitated multiplex gene expression studies. The substrate geissoschizine was produced as described previously[5], and _O_-acetylstemmadenine (OAS) was semi-purified from a _C. roseus_ mutant of _O_-acetylstemmadenine oxidase (ASO) gene, which accumulated almost exclusively OAS instead of catharanthine and vindoline[3]. Transient expression of ASO, DPAS, and HL2 in tobacco and OAS substrate feeding led to the expected formation of tabersonine (Supplementary Figure 6). In these reactions, reduction of OAP leads to the simultaneous OAP deacetylation and the formation of the ring-opened dehydrosecodine, which is further cyclized by HL2 to form tabersonine (Fig. 1) as shown in previous studies[3,7,30,31]. Next, the addition of VNS to the expression cocktail diverted the highly reactive dehydrosecodine intermediate towards the formation of 19_R_- and 19_S_-vindolinine, since the transient expression of ASO, DPAS, VNS and HL2 in tobacco and OAS substrate feeding led to the formation of 19_R_- and 19_S_-vindolinine instead of tabersonine (Supplementary Fig. 6). Surprisingly, a third product venalstonine formed in the feeding assays, which extended the VNS product spectrum. It is also worth noting that VNS activity was almost abolished when it was His-tagged at C-terminus (Supplementary Fig. 6).

**VNS catalyzes an unusual redox-neutral reaction to form vindolinines.** The recombinant VNS with N-terminal His-tag was expressed and purified from _E. coli_ for in vitro studies (Supplementary Fig. 7). The addition of OAP to DPAS and HL2 enzymes converted OAP to tabersonine, while addition of VNS to the incubation mixture led to the formation of 19 _S/R_-vindolinines and venalstonine, at the expense of tabersonine (Fig. 2d). Additional studies showed that DPAS could be substituted with GS, and HL2 could be substituted by other homologous hydrolases in _C. roseus_ such as HL4 and a hydrolase from _Vinca minor_ (VmHL) (Supplementary Fig. 8). In contrast, the hydrolase HL1 required for formation of catharanthine was inactive and could not replace HL2 or HL4 to produce 19 _S/R_-vindolinines or venalstonine, while HL3 showed much reduced activities compared to HL2 (Supplementary Fig. 8).

We then set out to determine the co-factor requirement for VNS. Similar to other known Fe/2OG dioxygenases, VNS in vitro activity was dependent on both α-ketoglutarate and ascorbate, while the addition of Fe(II) was able to more than double the enzyme activities but was not absolutely required (Table 1).

Fe/2OG dioxygenases are considered to adopt a similar, two-step catalysis mechanism that involves $O_2$ docking, α-ketoglutarate decarboxylation, and the generation of the reactive Fe(IV)-oxo species in the first half of the reaction[32,33]. In most cases, the Fe(IV) = O further abstract a hydrogen from the primary substrate, and the resulting radical rebounds to the Fe(III)-OH leading to substrate hydroxylation and Fe(III) reduction to regenerate Fe(II) (Fig. 3a). VNS reaction is highly unusual because the product does not contain a hydroxyl group and the overall reaction with respect to vindolinines is redox neutral. We suspect that VNS reaction follows the general reaction steps and forms a C19 radical (C18 radical for

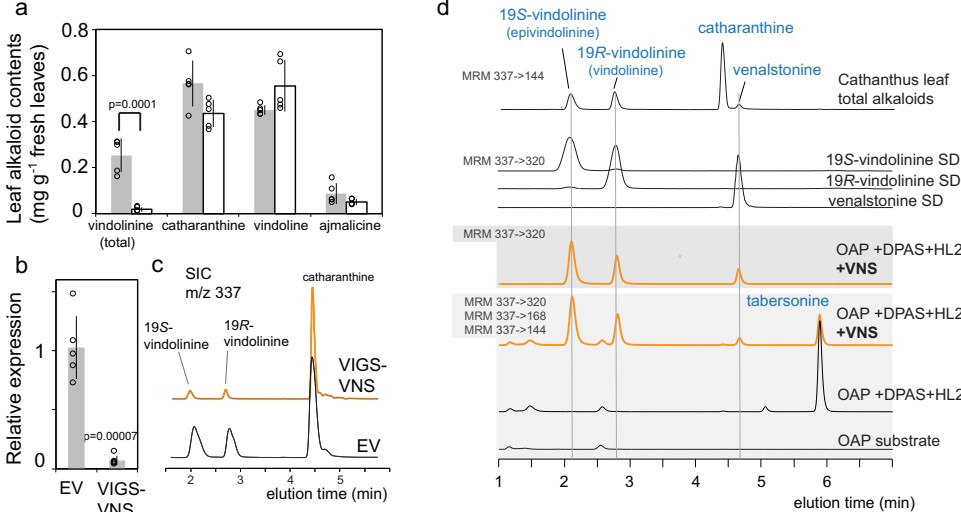

**Fig. 2 Vindolinine synthase (VNS) is responsible for the biosynthesis of 19 *S/R*-vindolinine and venalstonine in vivo and in vitro. a** Virus-induced gene silencing (VIGS) resulted in 95% reduction in the levels of the 19*R*-vindolinine and 19*S*-vindolinine in *C. roseus* leaf compared to the empty vector (EV) controls. **b** In the VIGS plants, the VNS transcripts were reduced by 93% compared to the EV controls. **c** Representative LC-MS chromatogram showing the reduction of vindolinine 19-epimers in VIGS plants ($m/z$ 337). **d** Tabersonine was biosynthesized when *O*-acetylprecondylocarpine (OAP) is incubated with dihydroprecondylocarpine acetate synthase (DPAS) plus Hydrolase 2 (HL2) recombinant enzymes. Addition of VNS to DPAS and HL2 recombinant enzymes converted OAP to 19*R*-vindolinine, 19*S*-vindolinine, and venalstonine at the expense of tabersonine. The following multiple reaction monitoring (MRM) ion transitions were used for detection: $m/z$ 337->320 (vindolinine 19-epimers and venalstonine); $m/z$ 337->144 (catharanthine, vindolinine 19-epimers, and venalstonine); $m/z$ 337->168 (tabersonine). The alkaloid MS spectra used for selecting these parameters are found in Supplementary figure 3. In VIGS experiments, the data was generated by taking the mean values from 5 VIGS and 5 EV control plants. The error bars indicate the standard deviation. Unpaired two-tailed Student's *t*-test was used for statistical analyses in VIGS experiments. For total vindolinine, the *p* value was 0.0001; for catharanthine, the *p* value was 0.0373; for vindoline, the *p* value was 0.0756; for ajmalicine, the *p* value was 0.110. The *p* value for relative VNS expression between EV and VIGS plants was 0.00007.

---

**Table 1 Vindolinine synthase (VNS) requires both α-ketoglutarate and ascorbate for activity in vitro.**

| α-keto-glutarate (mM) | Ascorbate (mM) | FeSO₄ (μM) | VNS pre-incubated with 2 μM FeSO4 | Relative activity (%) |
|---|---|---|---|---|
| 0 | 7.5 | 0 | YES | 0 |
| 0.2 | | | | 104 |
| 0.5 | | | | 100* |
| 2 | | | | 86 |
| 0.5 | 0 | 0 | YES | 0 |
| | 0.5 | | | 34 |
| | 7.5 | | | 100 |
| | 20 | | | 47 |
| 0.5 | 7.5 | 0 | NO | 59 |
| | | 2 | NO | 231 |
| | | 10 | NO | 249 |
| | | 2 | YES | 239 |

*Standard reaction condition used in this study. Relative activity is defined as the relative amounts of vindolinine epimers and venalstonine produced in other conditions when compared to the standard reaction in 1 hr reaction time.

---

venalstonine), which explains the rather equal production of the vindolinine 19-epimers. Instead of hydroxylating C19, the VNS Fe(III)-OH center may abstract the hydroxyl from water to generate $H_2O_2$, while the hydrogen from water bonds to C16 (Fig. 3b).

By replacing the $H_2O$ in the reaction with $D_2O$, we observed the incorporation of one deuterium from $D_2O$ into the vindolinines and venalstonine since the $m/z$ values of these three ($m/z$ 337) were all increased by 1 ($m/z$ 338) (Fig. 3c, Supplementary Fig. 9). The rather high percentage (23%, Fig. 3c) of $m/z$ 338 species in 100% $H_2O$ reaction is likely caused by the low mass resolution (0.7 amu) of the LC-MS/MS instrument used in this study. In comparison, the mass of tabersonine ($m/z$ 337) formed in $D_2O$ solution remained unchanged (Supplementary

Fig. 9), supporting the well-accepted cycloaddition mechanism and no involvement of water. To investigate the generation of $H_2O_2$ or •OH radical, we measured the formation of threonate after VNS reaction since ascorbate is readily oxidized by these species to a number of products including the more stable threonate[34]. While ascorbate was spontaneously and rapidly oxidized to threonate in aerobic solution even without VNS, we observed a statistically significant increase of threonate formation in VNS catalyzed reactions (Supplementary Fig. 9), suggesting that ascorbate is the final electron donor in VNS reaction.

We further investigated the VNS reaction without the cyclase HL2, and two further reduced, isomeric ($m/z$ 339) products were observed with only DPAS and VNS when reacted with OAP (Fig. 3d, Supplementary Fig. 10). To examine these two reduced

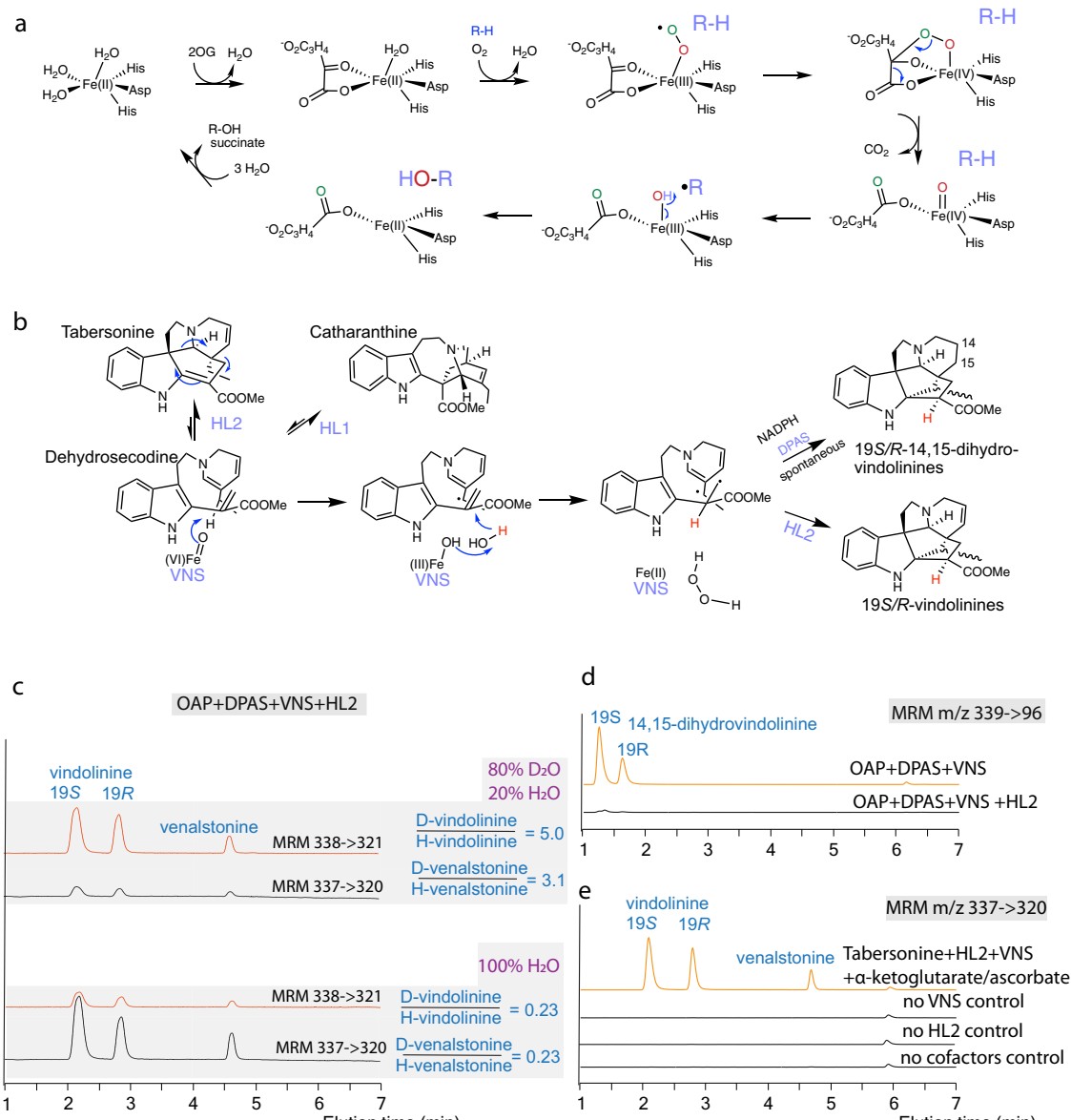

**Fig. 3 Vindolinine synthase (VNS) catalyzes an unusual, redox neutral reaction with respect to the principal substrate dehydrosecodine and the product vindolinine epimers. a** Typical reactions by Fe/2OG dioxygenases generate a substrate radical by enzyme-bound Fe(IV) = O, which rebounds with the hydroxyl radical leading to substrate hydroxylation. **b** Proposed VNS reaction involves the radical formation by Fe(IV) = O and water oxidation to regenerate Fe(II). The resulting di-radical is either cyclized by hydrolase 2 (HL2) to form the 19 S/R-vindolinines and venalstonine or spontaneously cyclizes with further reduction to form 19 S/R-14,15-dihydrovindolinines. **c** Replacing the reaction in 80% $D_2O$ resulted in the incorporation of one deuterium into the vindoline/venalstonine structures as the m/z values were all increased by 1 amu. **d** In the absence of HL2, the reaction product was further reduced by DPAS and spontaneously cyclized to form 19 S/R-14,15-dihydrovindolinines. **e** HL2 catalyzed the reverse reaction to generate dehydrosecodine from tabersonine, which was catalyzed by VNS and HL2 in the forward reaction to generate 19 R/19S-vindolinine epimers and venalstonine. The alkaloid MS spectra used for selecting MRM parameters are found in Supplementary figure 3 and Supplementary figure 9.

products, we hydrogenated the 14,15-double bonds of 19 S/R-vindolinines with palladium and $H_2$ and produced the 19 S/R-14,15-dihydrovindolinines, which turned out to be identical with the enzymatically produced m/z 339 species in LC-MS/MS (Supplementary Fig. 10).

The VNS-catalyzed isomerization reaction is further investigated using the reverse reaction, specifically by reacting (-)-tabersonine and (-)-vincadifformine (14,15-dihydrotabersonine) with HL2 and VNS. The results showed that HL2 was able to catalyze the reverse reaction to produce the ring-opened secodine type intermediates (Fig. 3b, e, Supplementary Fig. 11). The highly reactive intermediate dehydrosecodine and secodine

generated by HL2 were converted by the forward, VNS-HL2 mediated reaction to 19 S/R-vindolinines/venalstonine and 19 S/R-14,15-dihydrovindolinines respectively using (-)-tabersonine and (-)-vincadifformine (Fig. 3e, Supplementary Fig. 11). These results also suggested that the VNS reaction does not involve an initial oxidation followed by an enzyme-mediated reduction, because no reductase was presented. In addition, VNS also accepted the indole-hydroxylated substrate 11-hydroxy-dehydrosecodine, since the reaction of 11-hydroxytabersonine with HL2 and VNS produced the 19 S/R-11-hydroxyvindolinines and 11-hydroxyvenalstonine with expected mass, although the enzymes instead showed a strong preference for producing the 19S-epimer

(Supplementary Fig. 11). Using the same HL2/VNS reaction, no products were formed when the substrates were 19-hydroxyvincadifformine (minovincinine), 15-hydroxyvincadifformine, or 14,15-epoxytabersonine (lochnericine). However, it is not clear whether these non-indole substitutions are not accepted by HL2 or VNS, since HL2 reverse reaction precedes that of VNS. Finally, we repeated the HL2-catalyzed reverse-cycloaddition reaction of (-)-tabersonine by including HL1 in the reaction mixture, which resulted in the conversion of (-)-tabersonine to catharanthine (Fig. 3b, Supplementary Fig. 11). Albeit barely detected by LC-MS/MS, we were able to convert trace amounts of catharanthine to vindolinine epimers by reacting catharanthine with HL1, HL2, and VNS (Supplementary Fig. 11).

## Discussion

Fe/2OG dioxygenases are a large family of oxidases found in all three kingdoms of life. Some are well known for their critical roles in primary metabolism such as proline hydroxylases involved in collagen modification, histone demethylases involved in gene activation, and the oxidases in the biosynthesis of plant hormone gibberellins[32,33]. Several of them are also known in other specialized metabolism reactions, where oxidation and oxidative rearrangement is key to the diversification of chemical structures and their bioactivities. In these reactions, one oxygen atom of $O_2$ is inserted into the co-substrate α-ketoglutarate, and the other is usually inserted in the primary substrate resulting in a single hydroxylation. If the hydroxylation is on a carbon bound to a heteroatom (N/O), the hydroxylation usually leads to the demethylation/deacylation. In many cases, the Fe(III)-OH may also abstract a second hydrogen from an adjacent carbon/hydroxyl, and the resulting di-radical connect and form a double bond or a new ring on the substrate. Some Fe/2OG dioxygenases have further evolved to catalyze halogenation. In typical Fe/2OG dioxygenases, the iron is coordinated by a facial triad comprised of two histidines and an acidic Glu/Asp residue. However, in Fe/2OG halogenases, the acidic residue is replaced with an Ala/Gly residue, allowing the coordination of a large halogen anion, which results in substrate substitution with the halogen radical rather than the hydroxyl radical[35].

Among all known Fe/2OG dioxygenases with diverse activities, the redox-neutral reaction by VNS with respect to the substrate dehydrosecodine ($C_{21}H_{24}N_2O_2$) and the isomeric product vindolinines ($C_{21}H_{24}N_2O_2$) is highly unusual. There are only few redox-neutral reactions by Fe/2OG dioxygenases reported in nature. A well-studied example is the carbapenam synthase (CarC) catalyzed epimerization of (3 S,5 S)-carbapenam to (3 S,5 R)-carbapenam in the biosynthesis of several β-lactam antibiotics from the bacterium *Pectobacterium carotovorum*[36,37]. Specifically, the Fe(IV) = O in CarC abstracts a hydrogen from the substrate, then a tyrosine residue of CarC donates a hydrogen to the radical on the opposite plane of the substrate leading to the epimerization. In CarC, the Fe(III)-OH is not reduced during the reaction, therefore the enzyme is limited to a single turnover[37]. A second example is the AndA catalyzed rearrangement/epimerization in the biosynthesis of the triterpenoid Anditomin in the fugus *Aspergillus variecolor*[38]. In this reaction, AndA generates a substrate radical, which triggers a set of ring re-arrangements and eventually forms andiconin, an isomer of the substrate pre-andiloid C. It is worth noting that both the bacterial CarC and the fungal AndA are able to catalyze a regular oxidation reaction in the same biosynthetic pathways using a different substrate, which in both cases forms a double bond.

In *C. roseus*, *O*-acetylstemmadenine (OAS) is first oxidized to *O*-acetylprecondylocarpine (OAP). The following NADPH-dependent reduction leads to spontaneous deacetylation and ring-opening, forming dehydrosecodine and its iminium, which are cyclized by a number of α/β-hydrolases into various MIA skeletons (Fig. 1)[3,6,7,30,31]. VNS is not active against OAP or tabersonine. It is only active when OAP is reduced by DPAS or when tabersonine is reversely converted by HL2. This evidence suggest that the VNS substrate is most likely dehydrosecodine. When dehydrosecodine is cyclized directly by HL2, it forms the isomeric tabersonine as all studies have suggested so far. VNS reaction instead diverted this conversion to produce three isomeric, alternatively cyclized products without oxidation. VNS showed typical Fe/2OG dioxygenases motifs and does strictly require α-ketoglutarate, the co-substrate that receives an oxygen atom from splitting of $O_2$ in all known cases. It is surprising that the other oxygen atom is not incorporated in any MIA products, nor have the MIA products been oxidized in any way or lost a carbon because of this oxidation.

In the reverse reaction catalyzed by HL2 and VNS, vindolinines and venalstonine were formed without any reductase. The results suggest that VNS does not oxidize the MIA substrate, which is similar to the reactions catalyzed by CarC and AndA. A possible explanation for the destiny of the other oxygen atom of $O_2$ is that it is used to oxidize water to $H_2O_2$, which is reduced by ascorbate to generate threonate and water[32]. This is supported by the incorporation of a deuterium in the vindoline epimers and venalstonine from $D_2O$. It is also possible that ascorbate instead is oxidized by VNS's Fe(III)-OH to form dehydroascorbate, which spontaneously continues to be hydrolysed and oxidized to threonate. In this process, ascorbate donates a hydrogen to C16 of the vindolinine epimers and venalstonine, and a second hydrogen to Fe(III)-OH, which reduces it to Fe(II)-$H_2O$. Since ascorbate is in constant exchange of protons with water, this hypothesis can also explain the incorporation of a deuterium from $D_2O$. The absolute requirement of ascorbate in VNS reaction also supports this hypothesis. In both scenarios, ascorbate acts as the final electron donor and reductant, which is supported by the small but statistically significant increase of threonate production when VNS was involved (Supplementary Fig. 9). Nonetheless, the role of ascorbate in many other Fe/2OG dioxygenase reactions remains unclear[32,33], and we cannot rule out other possible reaction mechanisms.

It is interesting to observe the spontaneous cyclization of VNS product in the absence of the cyclase HL2. In this case, the 14,15-double bond is further reduced by DPAS activity to form 14,15-dihydrovindolinines. This agrees with the previous observation that dehydrosecodine iminium spontaneously cyclizes and gets reduced to vincadifformine in the absence of a hydrolase[30]. The formation of 14,15-dihydrovindolinines is therefore explained.

The vindolinine epimers are the third most abundant MIA in *C. roseus*, the model plants for plant specialized metabolism, after catharanthine and vindoline. Silencing VNS in *C. roseus* resulted in more than 95% reduction of vindolinines, which is a strong support for its role *in planta*. Vindolinines are only reported in three MIA producing species, *C. roseus*, *Vinca erecta* and *Melodinus balansae*. There are no publicly available transcriptomes of the latter two species, but we suspect that similar dioxygenases may be responsible for vindolinine biosynthesis in them. By searching in the National Center for Biotechnology Information (NCBI) total nonredundant protein database and the several sequenced transcriptomes of MIA producing species that have not been reported for vindolinines, we could not find a VNS homolog of more than 50% amino acid identity, which suggest that VNS evolved specifically within a few genera of Apocynaceae family. A further genomic investigation may answer whether other Apocynaceae plants encode a VNS homolog in the genome, which is likely not expressed or expressed with low levels in the plants due to the absence of vindolinines. Compared to the

restricted distribution of 19R-, and19S-vindolinines, derivatives of venalstonine are found in many more Apocenaceae species including the genera of *Kopsia, Melodinus, Pleiocarpa, Aspidosperma,* and *Alstonia*[25]. Based on VNS's ability to produce small amounts of venalstonine, it is possible that the biosynthesis of more wide-spread venalstonine derivatives involves similar Fe/2OG dioxygenases that catalyze this unusual, redox-neutral isomerization that is specific for the formation of C18 rather than C19 radicals.

## Methods

**Chemical standards and purifications from plant materials**. The 19R-vindolinine (vindolinine) standard was purchased from Sigma Aldrich. Both 19R-vindolinine and 19S-vindolinine (epivindolinine) and venalstonine were also purified from *C. roseus* leaf total alkaloids. Specifically, 50 g of *C. roseus* leaves were submerged in chloroform for 5 min. The alkaloids were extracted from chloroform by 1 M HCl, which was basified to pH 8 and further extracted with ethyl acetate to afford total alkaloids. The alkaloids were separated by thin layer chromatography (TLC, silica gel 60 G F254, Sigma Aldrich) and the solvent ethyl acetate: methanol 4:1 (v/v). The 19S-vindolinine (1 mg, Rf 0.32) and 19R-vindolinine (1 mg, Rf 0.36) were harvested from TLC. Venalstonine (Rf 0.61) contaminated with vindoline was further separated by TLC with pure methanol, which afforded less than 0.1 mg venalstonine (Rf 0.43). The remaining MIA standards used in this study were described previously[3–6,39].

**NMR and LC-MS/MS**. The NMR spectra was recorded using a Varian Unity 400 MHz spectrometer with CDCl₃ referenced at 7.26 ppm or acetone d6 at 2.05 ppm. The reference spectra of venalstonine were kindly provided by Prof. Kam Toh Seok at the University of Malaya, Malaysia to help identify venalstonine of low quantity. LC-MS/MS was performed on an Agilent Ultivo Triple Quadrupole LC-MS equipped with an Avantor® ACE® UltraCore™ SuperC18™ column (2.5 μm, 50x3mm), which included the solvent systems: solvent A, methanol: acetonitrile: ammonium acetate 1 M: water at 29:71:2:398; solvent B, methanol: acetonitrile: ammonium acetate 1 M: water at 130:320:0.25:49.7. The following linear gradient (8 min, 0.6 ml/min) were used: 0 min 80% A, 20% B; 0.5 min, 80% A, 20%B; 5.5 min 1% A, 99% B; 5.8 min 1% A, 99% B; 6.5 min 80% A, 20% B; 8 min 80% A, 20% B. The photodiode array detector records from 200 to 500 nm. The MS/MS was operated with gas temperature at 300 °C, gas flow of 10 L/min, capillary voltage 4 kV, fragmentor 135 V, collision energy 30 V with positive polarity. The Qualitative Analysis 10.0 software by Agilent was used for all LC analyses. The analytes were either dissolved in methanol or methanol: water in equal volume ratio. MIAs were identified and quantified using peak areas (UV 280 nm: catharanthine; UV 300 nm: vindolinine epimers, vindoline, and vindorosine; UV 330 nm: all aspidosperma MIAs) by comparing to serial dilutions of authentic standards. The MIA contents were calculated per fresh sample weight.

**Gene cloning**. VNS was amplified from *C. roseus* total cDNA with primers set (1/2) and (1/3). The PCR reactions generated VNS gene fragments with and without a stop codon, which is cloned into the tobacco expression vectors pEAQ-HT-DEST2 and -DEST3 for expression with a N-terminal Histag and a C-terminal Histag by Gateway® cloning according to manufacturer's protocol (Thermo Fisher). For VNS expression in *E. coli*, the gene was amplified by primer set (4/5) and subcloned into pET30b+ vector within BamHI/SalI sites. For silencing VNS in *C. roseus* by virus-induced gene silencing, a fragment of VNS was amplified by primer set (6/7) and cloned in pTRV2 vector within EcoRI site. The primers are listed in Supplementary table 3. GS (Genbank MF770507), ASO (Genbank MH136588), DPAS (Genbank KU865331), HL1–4 (Genbank MF770512-770515), VmHL (Genbank MH746436) genes used in this study were cloned previously[3–6]. pEAQ-HT-DEST2/3 vectors were mobilized to *Agrobacterium tumefaciens* (strain LBA4404). pET30b+ vector was mobilized to *E. coli* (stain BL21-DE3). pTRV2 vector was mobilized to *A. tumefaciens* (strain GV3101).

**Virus-induced gene silencing in *C. roseus***. VIGS experiment were performed as previously described[4] using *C. roseus* cv. Little Delicata seedlings. Overnight cultures (28 °C) of *A. tumorfaciens* (strain GV3101) cells harboring pTRV2-VNS, pTRV2-empty vector, pTRV2-CrPDS (phytoene desaturase), and pTRV1 vectors were harvested by centrifugation and resuspended in infiltration buffer (10 mM MES pH 5.6, 10 mM MgCl₂, 0.2 mM acetosyringone) to OD₆₀₀ = 1.5, which was cultured at 28 °C for 2.5 h. The suspensions of pTRV1 and pTRV2 were mixed, and a toothpick was dipped in the mixed suspension and used to penetrate the 4-week-old *C. roseus* seedlings just underneath the meristem. After penetration, an additional 0.12 ml suspension was used to flood the wound. Five seedlings were used to generate the biological replicates for each VIGS experiment. The infected seedlings were further cultured at lower temperature 20 °C (16/8 h photoperiod) in a greenhouse until the VIGS-PDS control seedlings started to show strong leaf bleaching (c.a. 4 weeks). The youngest pair of leaves, where silencing occurred, were harvested and split in half along the vertical main vain. One half of the split-leaf pair was used for RNA isolation using Trizol® reagent (Thermo Fisher, Waltham, USA) according to the manufacture's protocol and qRT-PCR studies as described previously[3–6]. The other half of the split leaves were submerged in 20 times (v/w) chloroform for 5 min. The chloroform extracts were removed and dried completely under vacuum, and were reconstituted in 20 times (v/w) methanol (leaf surface MIAs). The remaining leaf materials were submerged in 20 times (v/w) methanol for 1 h to generate leaf body MIAs. Filtered extracts (5 μL) were submitted to LC-MS/MS for MIA quantifications. The changes of MIA contents were evaluated by two-tailed, unpaired Student-t test from 5 independent biological replicates using Microsoft Excel.

**qRT-PCR**. qRT-PCR experiments were performed on an Agilent AriaMx Real-Time PCR instrument using the SensiFAST SYBR No-ROX qPCR 2X master mix (FroggaBio, Concord, Canada) according to the manufacture's protocol. The qRT-PCR (10 μL, 5 ng total RNA) cycles included 40 cycles of 95 °C for 10 s and 58 °C for 30 s. The Ct values and standard ΔΔCt method was used to quantify gene expression levels, which was normalized by using the expression of *C. roseus* 60 S ribosomal RNA housekeeping gene[3–6]. The changes of gene expression levels were evaluated by two-tailed, unpaired Student-t test from 5 independent biological samples with 3 technical replicates using Microsoft Excel.

**Epidermis-enriched leaf RNA isolation**. The carborundum abrasion method was used to isolate the epidermis-enriched leaf RNA as described[20] with modifications. *C. roseus* cv. Little Delicata leaves of approx. two centimeters in length (1 g) were collected in a 50 ml conical tube, then 1 g of carborundum (silicon carbide, 320 grit, Thermo Scientific, Waltham, USA) was added to the tube. After adding 3 ml RNA protection buffer from the Monarch® Total RNA extraction kit (New England Biolabs, Ipswich, USA), the mixture was vortexed for 30 s to release epidermis RNA. The remaining RNA isolation was performed according to the manufacture's protocol. The qRT-PCR primers are listed in Supplementary Table 3.

**Tobacco transient expression of multiple genes**. The tobacco transient expression experiments were performed as previously described[3]. Overnight culture (28 °C) cells harboring various pEAQ-HT-DEST vectors were harvested by centrifugation and resuspended in infiltration buffer (10 mM MES pH 5.6, 10 mM MgCl₂, 0.2 mM acetosyringone) to OD₆₀₀ = 1.5, which was cultured at 28 °C for 2.5 h. The suspensions for expressing different genes were mixed in equal volume and infiltrated to *Nicotiana benthamiana* leaves with a syringe. The tobacco plants were cultivated in a greenhouse at 26 °C (16/8 h photoperiod) for four days for protein expression and leaves were then infiltrated with 5–10 μg of alkaloid substrates in 10 mM Tris-HCl buffer at pH 8.0 and let react overnight. The leaves were extracted with methanol, and the extracts were used for LC-MS/MS analyses.

**Purification of recombinant VNS from *E. coli***. Overnight culture of *E. coli* strain BL21-DE3 harboring pET30b+VNS construct was used to inoculate 400 ml LB media shaking at 37 °C. When the culture OD₆₀₀ reached 0.3, the cells were induced with 1 mM IPTG at 37 °C for 2.5 h. The pelleted cells were sonicated in ice cold buffer (Tris HCl pH 7.5, 100 mM NaCl, 10% (v/v) glycerol), and the soluble proteins were obtained by centrifugation at 10,000 g for 10 min at 4 °C. VNS with N-terminal 6XHIStag was affinity purified by binding to a Ni-NTA resin and pure VNS was eluted with 250 mM imidazole in the same buffer. Finally, the proteins were desalted to remove imidazole using a PD10 column (GE Health Sciences) into the same buffer and stored at −80 °C.

**In vitro assay**. A typical in vitro assay (100 μL) included 50 mM HEPES buffer pH 7.5, 1 mM NADPH, 0.5 mM α-ketoglutarate, 7.5 mM ascorbate, 5 μg O-acetylprecondylocarpine (OAP) or other alkaloid substrates, and 2 μg each enzyme (DPAS, VNS, HLs). For GS, 50 μg proteins were used instead. FeSO4 (2–10 μM) may also be used. The reaction took place at 30 °C for 1 h, and stopped by adding equal volume of methanol. The reactions were further analyzed by LC-MS/MS. For all hydrolases-catalyzed reversed reactions, the reaction took place at 30 °C overnight. For reactions to determine threonate productions, the reactions (50 μL) included 20 mM Tris-HCl buffer pH 7.5, 1 mM NADPH, 0.5 mM α-ketoglutarate, 0.5 mM ascorbate, 5 μg O-acetylprecondylocarpine (OAP), 1 μg DPAS, 5 μg VNS, and 2 μg HL2. The reactions were started by adding DPAS and took place at 30 °C for 30 min, then they were terminated by mixing with 200 μL 75% (v/v) methanol.

**Vindolinine hydrogenation**. The purified 19S- and 19R-vindolinines (0.2 mg) were dissolved separately in 1 ml methanol in a small sample vial with trace amounts of Palladium on carbon 10 wt.% loading (Sigma Aldrich). The reaction was stirred at room temperature overnight under H₂ to afford the 14,15-dihydrovindolinine 19-epimers, respectively.

**Reporting summary**. Further information on research design is available in the Nature Research Reporting Summary linked to this article.

## Data availability

The raw data for Fig. 2a, b, Supplementary Fig. 4, 9e, and Supplementary table 2 are provided in the source data file. MS/MS and UV absorption data are provided in Supplementary fig.3. NMR data are provided in Supplementary fig. 1 and 2, and Supplementary table 1. Other data in this study are available from the corresponding author upon request. Source data are provided with this paper.

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

## Acknowledgements

We thank Prof. Kam Toh Seok at the University of Malaya, Malaysia for help in alkaloid identification. Y.Q. acknowledges the supports from the Natural Sciences and Engineering Research Council of Canada (NSERC) Discovery Grant, the Canada Foundation for Innovation John R. Evans Leaders Fund, and the New Brunswick Innovation Foundation grants RIF2019-036, EQP2020-007, RAI2021-068. V.D.L. acknowledges the support from the NSERC Discovery Grant. J.G.M.E. acknowledges the supports from the NSERC Undergraduate Student Research Awards (2020, 2021). M.S. acknowledges the stipend supports from the Department of Chemical Engineering at University of New Brunswick. V.D.L. was supported by a Canada Research Chair Tier1 grant. Y.Q. was supported by a Cannabis Health Research Chair grant from Tetra Bio Pharma Inc. and the New Brunswick Health Research Foundation.

## Author contributions

V.D.L. and Y.Q. conceived the research. J.G.M.E., M.S., D.P.S., and Y.Q. performed the experiments. J.H. contributed to chemical analyses and discussion. V.D.L. and Y.Q. wrote the manuscript.

## Competing interests

The authors declare no competing interests.
