## [Peer Review File · Nature Communications]

A *Catharanthus roseus* Fe(II)/ α -ketoglutarate-dependent dioxygenase catalyzes a redox-neutral reaction responsible for vindoline biosynthesisREVIEWER COMMENTS

Reviewer #1 (Remarks to the Author):

This manuscript authored by Qu et. al. described a new transformation in vindolinine biosynthetic pathway. Through transcriptomic analysis, in vivo and in vitro assays, an Fe/2OG dependent enzyme, vindolinine synthase (VNS) was identified to catalyze formation of 19S-vindolinine, 19R-vindolinine, and venalstonine. This discovery is of critical importance in which it solves the puzzle in the biosynthesis of these molecules. There is no doubt that the reaction catalyzed by VNS is novel. However, the mechanistic investigation raises a few questions. To help the reaction pathway elucidation, some experiments may need authors' attention.

In the proposed in vitro transformation of tabersonine (Figure 3B), the function of HL2 is to catalyze a reverse cycloaddition. Would it be possible to validate this hypothesis by following HL2 reaction?

Two radical species (mono- and di-radical) are proposed to be the key intermediates in the VNS catalyzed reaction. While diradical is very rare, the proposed pathway provides a plausible explanation. Introduction of a radical quencher to test and trap the radical will be helpful to test this hypothesis.

The authors proposed that hydrogen peroxide is the side product of the VNS catalyzed reaction. It would be worth monitoring the production of H₂O₂ during the reaction.

For D₂O assay, the result can be further clarified if the ratio of D-incorporated substrate and the non-deuterated substrate is reported.

Minor point:

Characterization of venalstonine purified from *C. roseus* can be improved. With crude ¹H-NMR and MS/MS, the possibility of misassignment cannot be completely ruled out. A cleaner ¹H-NMR and ¹³C-NMR spectra would be helpful.

Reviewer #2 (Remarks to the Author):

NCOMMS-21-49381

The medicinal plant *Catharanthus roseus* synthesizes an amazing diversity of bioactive terpene indole alkaloid compounds. The biosynthesis of some quantitatively important ones remained unresolved. Based on educated mining into the existing transcriptomic data, the authors of this manuscript identify a potential candidate enzyme likely to be responsible for the last biosynthetic step of the third most abundant MIAs in *C. roseus*, vindolinines. Using VIGS in *C. roseus*, transient expression in *Nicotiana benthamiana*, and recombinant enzyme produced in *E. coli*, they convincingly demonstrate the involvement of this Fe(II)/ α -ketoglutarate-dependent dioxygenase (coined vindolidine synthase, VNS) in the formation of the vindolinine 19R and 19S diastereoisomers and of an additional compound, venalstonine. Using the recombinant enzyme, they demonstrate the α -ketoglutarate and ascorbate co-factor requirement and Fe(II) enhancement of VNS. Then they seek to explain the very unusual redox-neutral reaction catalyzed by the enzyme, which does not seem to involve any substrate oxidation. Based on additional experiments, they propose two potential chemical mechanisms where oxygen is reduced to H₂O₂ or ascorbate oxidized by VNS. The experiments are informative. They support the hypothesis that VNS does not oxidize its substrate and show that the substrate incorporates deuterium from D₂O during the reaction. Yet, the reaction mechanisms could not be unambiguously solved. Results are discussed in terms of reaction mechanism and pathway/gene

evolution in Apocynaceae, pointing to the unique catalytic properties of the enzyme and its potential role in other related plant species.

The manuscript is clear, concise and well-written. It just deserves some minor improvements:

- Results: the C-terminus of VNS by itself might not be important for catalysis but the introduction of the His tag could result in steric hindrance preventing access of substrate, co-factor or interacting protein, among other possibilities. It would thus be better to rephrase this section.
- In the discussion line 15 end of the line "...halogen anion, which is resulting..."
- In the discussion last page line 7 "...incorporation of a deuterium in the vindolinine epimers...".
- Figures 1 would be improved with carbon numbering of the main common substrate.
- Figures 1 and 3A are too small to be readable.
- LC profiles would deserve an X axis.
- In figure 3 legend, line 6 "to regenerated", line 8 "spontaneously cyclizes".
- The resolution and annotation of the tree in Sup figure 4 could be improved. Enzymes of Apocynaceae could be indicated (if any included in the tree).

Reviewer #3 (Remarks to the Author):

The manuscript of Ga Lay Eng and coworkers describes the identification of a new Fe/2OG dioxygenase involved in the synthesis of monoterpene indole alkaloids (MIAs) in the Madagascar periwinkle. While almost all the enzymes involved in the synthesis of the main MIAs produced in this plant have been identified over the last few years, some critical steps remained to be elucidated. In line, the authors characterized the molecular/enzymatic mechanisms involved in the synthesis of three cyclized alternative MIAs (19S and 19R vindolinine and venalstonine). This identification is of importance since the sum of vindolinine epimers is the third most abundant "MIA" accumulated in *Catharanthus roseus*. Interestingly, the dioxygenase catalyzing this synthesis (named vindolinine synthase) divert the biosynthetic route to tabersonine by metabolizing the dehydrosecodine generated by DPAS to from an alternative substrate to HL2 (tabersonine synthase) resulting in the formation of the vindolinine type MIAs. Such a molecular mechanism agrees with the detection of large amounts of vindolinine synthase transcripts in the leaf epidermis that hosts the aforementioned reactions. It is also worth mentioning that this vindolinine synthase corresponds to the first plant Fe/2OG dioxygenase catalyzing a redox-neutral reaction discovered to date. Experiments have been well conducted, the results are convincing, and the manuscript is well written. A few concerns are listed below:

- To conduct MIA quantification, the authors used leaf chloroform dipping to selectively extract MIAs. While this technique can provide interesting results, it is also controversial as reported in distinct previous articles. To remove any ambiguity, I suggest confirming their quantification by a second approach of their choice.
- Authors showed that vindolinine synthase transcripts are mainly accumulated in leaf epidermis by analyzing their epidermome database. This is an interesting result, but it does not provide any other information regarding expression in other tissue types. Do complementary experiments can be envisaged to provide this information (in situ Hybridization RNA, microcapture laser as authors already did or qPCR comparing epidermis vs the whole leaf).
- Studying the general expression profile of vindolinine synthase in the whole set of organs in comparison with MIA accumulation would be also very informative.
- What about the genomic environment of vindolinine synthase gene? Can a comparison with D4H and/or other dioxygenase can be envisaged?
- The VIGS approach provided interesting results that deserves to be validated by a statistical analysis. Did the author also try overexpression of this gene in the whole plant?
- Supplemental table 2: ajmalicine/serpentine are heteroyohimbine and not per se pre-stemmadenine

MIA, rather MIAs distinct from stemmadenine derived MIAs
- Supplemental Fig 4B. please highlight D4H to facilitate reading

REVIEWER COMMENTS

Reviewer #1 (Remarks to the Author):

This manuscript authored by Qu et. al. described a new transformation in vindolinine biosynthetic pathway. Through transcriptomic analysis, in vivo and in vitro assays, an Fe/2OG dependent enzyme, vindolinine synthase (VNS) was identified to catalyze formation of 19S-vindolinine, 19R-vindolinine, and venalstonine. This discovery is of critical importance in which it solves the puzzle in the biosynthesis of these molecules. There is no doubt that the reaction catalyzed by VNS is novel. However, the mechanistic investigation raises a few questions. To help the reaction pathway elucidation, some experiments may need authors' attention.

We thank the reviewer for the compliments and suggestions. Please find our responses below, and all changes made in the manuscript and supplementary materials are marked in red colour.

In the proposed in vitro transformation of tabersonine (Figure 3B), the function of HL2 is to catalyze a reverse cycloaddition. Would it be possible to validate this hypothesis by following HL2 reaction?

We thank the reviewer for the suggestion. The reverse reaction catalyzed by HL2 was indeed interesting. Previous work from our group and Prof. O'Connor's group both showed that the forward reaction includes generating a highly reactive and unstable dehydrosecodine intermediate that does not survive any formal analyses. The article by the O'Connor group (Caputi et al., 2020, *Nat. Chem. Bio*) showed that dehydrosecodine intermediate under their tested condition is also spontaneously cyclized to several by-products including a solvent protected MIA intermediate, proposed as "angryline", which may be converted back to dehydrosecodine at high pH 9.5 and therefor serves as the substrate for HL1 and HL2. Based on the MS/MS patten, we are also able to observe "angryline" or a MIA very similar to "angryline", but more analyses on these highly unstable intermediates are challenging.

Instead, we incubated tabersonine substrate with both HL2 and HL1 (catharanthine synthase), and we are able to show clear detection of catharanthine formation, which is not present when HL1 (catharanthine synthase) is absent. We hope this new data is another support for HL2-catalyzed reverse cycloaddition to generate the highly unstable dehydrosecodine intermediate, which is converted by HL1 to catharanthine. The new data is included in supplementary figure 11e. In the reverse reactions, we could not detect "angryline" or any other MIAs, likely because the reverse reaction is rather slow (overnight reaction). The small amounts of intermediates may not survive before analyses or they were immediately consumed by VNS or HL1 to the more stable products. To a much smaller extent, we were also able to show the reverse reaction of HL1, by incubating catharanthine with HL1, HL2 and VNS, which produced both vindolinine epimers (supplementary fig. 11f). We have included these two experiments in the manuscript.

Two radical species (mono- and di-radical) are proposed to be the key intermediates in the VNS catalyzed reaction. While diradical is very rare, the proposed pathway provides a plausible explanation. Introduction of a radical quencher to test and trap the radical will be helpful to test this hypothesis.

We agree with the reviewer that the di-radical is rare, and we would like to only propose it as one of the possible reaction mechanisms. In our dioxygenase reaction, there are high concentrations (0.5-7.5 mM) of ascorbate (vitamin C), which is not only a co-factor for dioxygenases but also a radical quencher. The role of ascorbate in many dioxygenase reactions is likely resulted from its reducing/quenching ability. Our VNS is not active in the absence of ascorbate, and is only active with rather high amounts of it (>0.5 mM ascorbate). The biological role of ascorbate includes quenching/reducing reactive oxygen species and other radicals generated during cell metabolism, with concomitant ascorbate oxidation into a number of products (Dewhirst and Fry, 2018, *Biochem J*). In all our VNS reactions in the presence of ascorbate, we did not observe major by-products or intermediates related with vindolinines (or anything that could be detected by our LC-MS/MS method), which could suggest that the radical species is channeled quickly from VNS to HL2 for cyclization or other mechanism is involved.

The authors proposed that hydrogen peroxide is the side product of the VNS catalyzed reaction. It would be worth monitoring the production of H₂O₂ during the reaction.

We thank the reviewer for the suggestion. We reasoned that if H₂O₂ or OH-radical were to be produced during the reaction, they would react quickly with the large amounts of ascorbate. We also proposed that ascorbate may be directly involved in reducing Fe³⁺ to Fe²⁺ during the reaction. In both scenarios, ascorbate is the final electron donor, and water is produced as an end product. As a result, we tried to quantify ascorbate oxidation since H₂O₂ or OH-radical quickly react with ascorbate. Ascorbate (M-H- m/z 175) oxidation is a complex process involving multiple oxidation routes, and one of the stable end products is threonic acid (M-H- m/z 135, figure below) that can be

detected in our LC-MS system (Dewhirst and Fry, 2018, Biochem J). Ascorbate also spontaneously and quickly oxidizes in aerobic aqueous solution into threonic acids and other products (Dewhirst and Fry, 2018, Biochem J).

In an effort to measure threonic acid production, we were able to show a statistically significant ($P=0.0448$) difference in the production of threonic acid in reactions with VNS and without VNS after 30 min of reaction (Figure below), which is also included in the Supplementary fig. 9d/e. Threonic acid is already presented in freshly prepared ascorbate, and its content increases rapidly due to O₂ oxidation even without the dioxygenase (complete oxidation occurs overnight spontaneously). However, in the reaction with VNS, we observed slightly higher but statistically significant production of threonic acid. While ascorbate oxidation does not unambiguously suggest production of H₂O₂, we hope this piece of information could support the reductive role of ascorbate in VNS reaction, either directly or indirectly. We have also updated the discussion on the role of ascorbate in the discussion section:

“A possible explanation for the destiny of the other oxygen atom of O₂ is that it is used to oxidize water to H₂O₂, which is reduced by ascorbate to generate threonate and water³². This is supported by the incorporation of a deuterium in the vindoline epimers and venalstonine from D₂O. It is also possible that ascorbate instead is oxidized by VNS’s Fe(III)-OH to form dehydroascorbate, which spontaneously continues to be hydrolysed and oxidized to threonate. In this process, ascorbate donates a hydrogen to C16 of the vindolinine epimers and venalstonine, and a second hydrogen to Fe(III)-OH, which reduces it to Fe(II)-H₂O. Since ascorbate is in constant exchange of protons with water, this hypothesis can also explain the incorporation of a deuterium from D₂O. The absolute requirement of ascorbate in VNS reaction also supports this hypothesis. In both scenarios, ascorbate acts as the final electron donor and reductant, which is supported by the small but statistically significant increase of threonate production when VNS was presented (Supplementary fig. 9). Nonetheless, the role of ascorbate in many other Fe/2OG dioxygenase reactions remains unclear^{32,33}, and we cannot rule out other possible reaction mechanisms.”

For D₂O assay, the result can be further clarified if the ratio of D-incorporated substrate and the non-deuterated substrate is reported.

We thank the reviewer again for the suggestion. The ratio has been included in Figure 3c

Minor point:

Characterization of venalstonine purified from *C. roseus* can be improved. With crude ¹H-NMR and MS/MS, the possibility of misassignment cannot be completely ruled out. A cleaner ¹H-NMR and ¹³C-NMR spectra would be helpful.

We have cleaned up our purified venalstonine by TLC. The slightly improved ¹H spectra are included in updated Supplementary fig. 2. We could not produce useful ¹³C-NMR due to the small quantity of venalstonine (<0.1 mg), because it is a minor alkaloid in *C. roseus*. We also enlarged the spectra from 1-4 ppm, to show the identical chemical shifts of our purified venalstonine and the reference venalstonine from Prof. Kam Toh Seok.

Reviewer #2 (Remarks to the Author):

NCOMMS-21-49381

The medicinal plant *Catharanthus roseus* synthesizes an amazing diversity of bioactive terpene indole alkaloid compounds. The biosynthesis of some quantitatively important ones remained unresolved. Based on educated mining into the existing transcriptomic data, the authors of this manuscript identify a potential candidate enzyme likely to be responsible for the last biosynthetic step of the third most abundant MIAs in *C. roseus*, vindolinines. Using VIGS in *C. roseus*, transient expression in *Nicotiana benthamiana*, and recombinant enzyme produced in *E. coli*, they convincingly demonstrate the involvement of this Fe(II)/ α -ketoglutarate-dependent dioxygenase (coined vindolidine synthase, VNS) in the formation of the vindolinine 19R and 19S diastereoisomers and of an additional compound, venalstonine. Using the recombinant enzyme, they demonstrate the α -ketoglutarate and ascorbate co-factor requirement and Fe(II) enhancement of VNS. Then they seek to explain the very unusual redox-neutral reaction catalyzed by the enzyme, which does not seem to involve any substrate oxidation. Based on additional experiments, they propose two potential chemical mechanisms where oxygen is reduced to H₂O₂ or ascorbate oxidized by VNS. The experiments are informative. They support the hypothesis that VNS does not oxidize its substrate and show that the substrate incorporates deuterium from D₂O during the reaction. Yet, the reaction mechanisms could not be unambiguously solved. Results are discussed in terms of reaction mechanism and pathway/gene evolution in Apocynaceae, pointing to the unique catalytic properties of the enzyme and its potential role in other related plant species.

The manuscript is clear, concise and well-written. It just deserves some minor improvements:

We thank the reviewer for the compliments and suggestions. Please find our responses below, and all changes made in the manuscript and supplementary materials are marked in red colour.

- Results: the C-terminus of VNS by itself might not be important for catalysis but the introduction of the His tag could result in steric hindrance preventing access of substrate, co-factor or interacting protein, among other possibilities. It would thus be better to rephrase this section.

Indeed, we have corrected the expression to "It is also worth noting that VNS activity was almost abolished when it was His-tagged at C-terminus" to avoid any confusions.

- In the discussion line 15 end of the line "...halogen anion, which is resulting..."

We have corrected the sentence accordingly.

- In the discussion last page line 7 "...incorporation of a deuterium in the vindolinine epimers..."

We have corrected the sentence accordingly.

- Figures 1 would be improved with carbon numbering of the main common substrate.

We have corrected the figure accordingly.

- Figures 1 and 3A are too small to be readable.

We have enlarged the figure elements to make them more visible.

- LC profiles would deserve an X axis.

We have included vertical lines in Figure 2, to align the peaks with their elution times (shared X-axis)

- In figure 3 legend, line 6 "to regenerated", line 8 "spontaneously cyclizes".

We have corrected the sentence accordingly.

- The resolution and annotation of the tree in Sup figure 4 could be improved. Enzymes of Apocynaceae could be indicated (if any included in the tree).

We have enlarged the figure elements to make them more visible. We have also marked all Apocynaceae dioxygenases in the phylogeny orange in colour.

Reviewer #3 (Remarks to the Author):

The manuscript of Ga Lay Eng and coworkers describes the identification of a new Fe/2OG dioxygenase involved in

the synthesis of monoterpene indole alkaloids (MIAs) in the Madagascar periwinkle. While almost all the enzymes involved in the synthesis of the main MIAs produced in this plant have been identified over the last few years, some critical steps remained to be elucidated. In line, the authors characterized the molecular/enzymatic mechanisms involved in the synthesis of three cyclized alternative MIAs (19S and 19R vindolinine and venalstonine). This identification is of importance since the sum of vindolinine epimers is the third most abundant "MIA" accumulated in *Catharanthus roseus*. Interestingly, the dioxygenase catalyzing this synthesis (named vindolinine synthase) divert the biosynthetic route to tabersonine by metabolizing the dehydrosecodine generated by DPAS to from an alternative substrate to HL2 (tabersonine synthase) resulting in the formation of the vindolinine type MIAs. Such a molecular mechanism agrees with the detection of large amounts of vindolinine synthase transcripts in the leaf epidermis that hosts the aforementioned reactions. It is also worth mentioning that this vindolinine synthase corresponds to the first plant Fe/2OG dioxygenase catalyzing a redox-neutral reaction discovered to date. Experiments have been well conducted, the results are convincing, and the manuscript is well written.

We thank the reviewer for the compliments and suggestions. Please find our responses below, and all changes made in the manuscript and supplementary materials are marked in red colour.

A few concerns are listed below:

- To conduct MIA quantification, the authors used leaf chloroform dipping to selectively extract MIAs. While this technique can provide interesting results, it is also controversial as reported in distinct previous articles. To remove any ambiguity, I suggest confirming their quantification by a second approach of their choice.

We thank the reviewer for the suggestion. We used the chloroform dipping method to purify vindolinine epimers for its effectiveness in removing vindoline and serpentine. For the leaf/flower alkaloid quantifications, they were done by grinding and extracting whole tissues with methanol. We have added one sentence in the manuscript to avoid any confusions about method used:

By extracting whole ground tissues with methanol, the major MIAs in *C. roseus* leaves and flowers were quantified using a number of standards in the plant variety "Little Delicata" and in another popular variety "Pacifica White". The results showed that while the vindolinine epimers were the third-most abundant MIAs in leaves of both varieties next to vindoline and catharanthine, they were the most abundant MIAs in flowers of Little Delicata and the second most abundant after vindoline/vindorosine in flowers of Pacifica White (Supplementary Table 2).

- Authors showed that vindolinine synthase transcripts are mainly accumulated in leaf epidermis by analyzing their epidermome database. This is an interesting result, but it does not provide any other information regarding expression in other tissue types. Do complementary experiments can be envisaged to provide this information (in situ Hybridization RNA, microcapture laser as authors already did or qPCR comparing epidermis vs the whole leaf).

We thank the reviewer for the suggestion. We have generated leaf epidermis enriched RNA by modified carborundum abrasion technique (included in the materials and methods section). We performed qRT-PCR to evaluate our epidermis enriched RNA, using three other MIA biosynthetic genes previously known to be expressed in different leaf cell types: geissoschizine oxidase (GO) in leaf epidermis, iridoid synthase (IS) in leaf IPAP cells, and deacetoxyvindoline 4-hydroxylase (D4H) in idioblast/laticifer cells. The results showed that the epidermis enriched RNA pool was effectively generated by carborundum abrasion method, as GO expression in these three individually prepared RNA samples were 1.7-fold higher, while IS and D4H expression were at 32% and 66% in these RNA sets, when compared to the whole leaf tissue (Figure below). Using these RNA sets, we showed that VNS expression is 3.3 time higher in epidermis-enriched leaf RNA compared to the whole leaf. This new data is included in Supplementary fig 4a, and we hope it serves a second piece of support for VNS subcellular localization. The unpaired student T-test was used for statistical analysis, which showed that the gene expression differences using three biological replicates and three technical replicates are highly statistically different ($P < 0.001$).

Supplementary figure 4. The transcripts of VNS were higher in the epidermis-enriched leaf total RNAs compared to the whole leaf total RNA (a). The expression of four MIA biosynthetic genes were tested by qRT-PCR in epidermis-enriched leaf total RNA prepared using carborundum abrasion and in whole *C. roseus* leaf. The epidermal MIA gene GO also showed higher transcripts in epidermis-enriched leaf total RNA, while the internal phloem associated parenchyma cell (IPAP)-localized iridoid synthase (IS) and laticifer/idioblast-localized deacetoxyvindoline 4-hydroxylase (D4H) showed reduced expression in these RNA sets. The unpaired two-tailed Student's *t*-tests were used to evaluate the differences between the gene expression in the whole leaf and epidermis-enriched leaf RNA for each gene. *** indicates p-value <0.001 in Student's *t*-test. (b) VNS expression in four *C. roseus* tissues: leaf, flower, root, and stem by qRT-PCR. Three independently prepared leaf epidermis-enriched RNA extracts were used in these studies. For each tissue type, three independently prepared RNA extracts were used. The gene 16S ribosome RNA was used to normalize each RNA sets. The error bars indicate standard deviations from three biological replicates and three qRT-PCR technical replicates.

- Studying the general expression profile of vindolinine synthase in the whole set of organs in comparison with MIA accumulation would be also very informative.

We thank the reviewer for the suggestion. We have included the qRT-PCR based gene expression levels of VNS in four different tissues: leaf, flower, root, and stem. The results showed that VNS is expressed in all these four tissues at various abundance. The information is included in Supplementary fig 4b (figure above).

- What about the genomic environment of vindolinine synthase gene? Can a comparison with D4H and/or other dioxygenase can be envisaged?

We thank the reviewer for the suggestion. We looked at the genomic loci for D4H and VNS using the only genomic source "Medicinal Plant Genomics Resources" at <http://mpgr.uga.edu>.

VNS is located on scaffold 3017763 (21.23kb length), and no other genes are annotated on this scaffold. D4H is located on scaffold 2969470 (16.04kb), no other genes are annotated on this scaffold. From the available data, we could not conclude the genomic linkage between these two genes due to the small scaffold sizes. Also VNS is only 30% identical to D4H at AA level. It is unlikely VNS is evolved from a gene duplication event from D4H, considering the low % identity. The VNS homolog with highest identity is found in *Vinca minor* (48% AA identity) as indicated in the manuscript. The phylogenetic tree in supplementary figure 5 included 27 Apocynaceae dioxygenases and many more from other plant species. We hope this serves good ground for the relatedness of VNS with other dioxygenases.

- The VIGS approach provided interesting results that deserves to be validated by a statistical analysis. Did the author also try overexpression of this gene in the whole plant?

We thank the reviewer for the suggestion. We have performed statistics using the unpaired two tailed Student's *t*-tests. While doing these tests, we discovered a previous mistake for catharanthine quantification in Figure 2a.

Previously, the catharanthine content in control EV plants was shown as 0.77 mg/g tissue due to an error in our excel spreadsheet. This has been corrected to 0.56 mg/g tissue. There are no other mistakes in any other quantifications after we carefully inspected all our data. The data used to generate all graphs has been included in the "Source Data" file. With the statistically tests, both the vindolinine contents (Fig. 2a) and VNS expressions (Fig. 2b) are statistically different ($P < 0.001$) between VIGS plants and EV plants. The updated figure is shown below.

We agree with the reviewer that over expressing VNS would provide more data on VNS function. While methods have been established for overexpressing genes in *C. roseus* hairy root cultures, there has not been a reliable method for overexpressing genes in whole *C. roseus*, or re-generate whole plants from transformed hairy roots after decades of research on this plant. Establishing a *C. roseus* transformation system may require several years and extensive experience in tissue culture. We hope that the virus induced gene silencing data, in vivo tobacco pathway reconstitution, qRT-PCR, in vitro enzyme assays, and other analyses in this manuscript are sufficient in supporting the biochemistry of VNS in *C. roseus* and its unusual redox-neutral catalytic capacity.

Figure 2. Vindolinine synthase (VNS) is responsible for the biosynthesis of 19S/R-vindolinine and venalstonine in vivo and in vitro. (a) Virus induced gene silencing (VIGS) resulted in 95% reduction in the levels of the 19R-vindolinine and 19S-vindolinine in *C. roseus* leaf compared to the empty vector (EV) controls. (b) In the VIGS plant, the VNS transcripts were reduced by 93% compared to the EV controls. (c) Representative LC-MS chromatogram showing the reduction of vindolinine 19-epimers in VIGS plants (m/z 337). (d) Tabersonine is biosynthesized when *O*-acetylprecondylocarpine (OAP) is incubated with dihydroprecondylocarpine acetate synthase (DPAS) plus Hydrolase 2 (HL2) recombinant enzymes. Addition of VNS to DPAS and HL2 recombinant enzymes convert OAP to 19R-vindolinine, 19S-vindolinine, and venalstonine at the expense of tabersonine. The following multiple reaction monitoring (MRM) ion transitions were used for detection: m/z 337->320 (vindolinine 19-epimers and venalstonine); m/z 337->144 (catharanthine, vindolinine 19-epimers, and venalstonine); m/z 337->168 (tabersonine). The alkaloid MS spectra used for selecting these parameters are found in Supplementary figure 3. In VIGS experiment, the data was generated by taking the mean values from 5 VIGS and 5 EV control plants. The error bars indicate the standard deviation. Unpaired two-tailed Student's *t*-test was used for statistical analyses in VIGS experiments. For total vindolinine, the P-value was 0.0001 (***); for catharanthine, the P-value was 0.0373 (*); for vindoline, the P-value was 0.0746; for ajmalicine, the p-value was 0.110. The P-value for relative VNS expression between EV and VIGS plants was <0.0001 (***).

- Supplemental table 2: ajmalicine/serpentine are heteroyohimbine and not per se pre-stemmadenine MIAs, rather MIAs distinct from stemmadenine derived MIAs

We thank the reviewer for the suggestion. We have changed the “pre-stemmadenine MIAs” to “other major alkaloids” in Supplementary table 2.

- Supplemental Fig 4B. please highlight D4H to facilitate reading

We thank the reviewer for the suggestion. D4H has been labeled in the phylogeny. We have also marked all Apocynaceae dioxygenases used in this analysis in orange colour for better presentation.

REVIEWERS' COMMENTS

Reviewer #1 (Remarks to the Author):

I appreciated the author's effort trying to address my questions. While there are still some questions that remained to be solved, this reviewer understands the practical challenges associated with those experiments and hopes that authors can continue exploiting the details of this interesting and uncommon reaction. With the revised draft, this reviewer supports the publication of this exciting story.

Reviewer #3 (Remarks to the Author):

The authors successively addressed the concerns raised by my previous reviews. This is a nice and very interesting story and I'm glad to see that enzymes catalyzing the synthesis of some main C. roseus MIAs can still be identified. Experiments and methodology are sound. As such, this manuscript is thus of interest for the whole community working on alkaloid metabolism but also on specialized metabolisms in general. The unusual reaction catalyzed by the dioxygenase makes this article also interesting for a broader community of researchers.